# Associations between Pre-Diagnostic Physical Activity with Breast Cancer Characteristics and Survival

**DOI:** 10.3390/cancers14071756

**Published:** 2022-03-30

**Authors:** Zi Lin Lim, Geok Hoon Lim, Peh Joo Ho, Alexis Jiaying Khng, Yen Shing Yeoh, Amanda Tse Woon Ong, Benita Kiat Tee Tan, Ern Yu Tan, Su-Ming Tan, Veronique Kiak-Mien Tan, Jingmei Li, Mikael Hartman

**Affiliations:** 1Laboratory of Women’s Health & Genetics, Genome Institute of Singapore, 60 Biopolis Street, Genome, #02-01, Singapore 138672, Singapore; lim_zi_lin@gis.a-star.edu.sg (Z.L.L.); hopj@gis.a-star.edu.sg (P.J.H.); khngja@gis.a-star.edu.sg (A.J.K.); 2Breast Department, KK Women’s and Children’s Hospital, Singapore 229899, Singapore; lim.gh@singhealth.com.sg; 3Duke-NUS Medical School, Singapore 169857, Singapore; 4Saw Swee Hock School of Public Health, National University of Singapore, Singapore 117549, Singapore; ephbamh@nus.edu.sg; 5Department of Surgery, Yong Loo Lin School of Medicine, National University of Singapore, Singapore 117597, Singapore; ysyeoh@nus.edu.sg; 6Department of Surgery, National University Hospital, Singapore 119054, Singapore; amanda_ong@nuhs.edu.sg; 7Department of Breast Surgery, Singapore General Hospital, Singapore 168753, Singapore; benita.tan.k.t@singhealth.com.sg (B.K.T.T.); veronique.tan.k.m@singhealth.com.sg (V.K.-M.T.); 8Department of General Surgery, Sengkang General Hospital, Singapore 544886, Singapore; 9Division of Surgery and Surgical Oncology, National Cancer Centre Singapore, Singapore 169610, Singapore; 10Department of General Surgery, Tan Tock Seng Hospital, Singapore 308433, Singapore; ern_yu_tan@ttsh.com.sg; 11Division of Breast Surgery, Changi General Hospital, Singapore 529889, Singapore; tan.su.ming@singhealth.com.sg

**Keywords:** breast cancer, physical activity, retrospective cohort study, cancer survival

## Abstract

**Simple Summary:**

Physical activity is known to reduce breast cancer risk and improve patient prognosis. However, the association between pre-diagnostic physical activity and the aggressiveness of breast cancer is unclear. Here, we assessed the effects of pre-diagnostic physical activity on breast cancer aggressiveness among breast cancer patients. Despite not improving overall survival, higher levels of pre-diagnostic physical activity contributed to less aggressive forms of breast tumours. These results underline the importance of physical activity in improving patient prognosis.

**Abstract:**

Physical activity (PA) is known to reduce breast cancer (BC) risk and improve patient prognosis. However, the association between pre-diagnostic PA and the aggressiveness of BC is unclear. We investigated the associations between PA, BC tumour characteristics, and survival. This retrospective observational study included 7688 BC patients from the Singapore Breast Cancer Cohort (2010–2016). PA information from the questionnaire included intensity (light/moderate/vigorous) and duration (<1 h/1–2 h/>2 h per week). A PA score (1–5) incorporating intensity and duration was calculated. Associations between PA score and tumour characteristics such as stage, histological grade, nodal and hormone receptor status were examined using multinomial regression. Moreover, 10-year overall survival was estimated using Cox regression analysis in 6572 patients after excluding patients with invalid survival data and stage IV disease. Breast tumours associated with higher PA score were more likely to be non-invasive (OR_invasive_ _vs._ _non-invasive(reference)_ [95% CI]: 0.71 [0.58–0.87], *p*-trend = 0.001), of lower grade (OR_poorly_ _vs._ _well differentiated(reference)_: 0.69 [0.52–0.93], *p* = 0.014), ER-positive (OR_ER-negative_ _vs._ _ER-positive(reference)_: 0.94 [0.89–1.00], *p*-trend = 0.049), PR-positive (OR_PR-negative_ _vs._ _PR-positive(reference)_: 0.82 [0.67–0.99], *p* = 0.041), HER2-negative (OR_HER2-negative_ _vs._ _HER2-positive(reference)_: 1.29 [1.02–1.62], *p*-trend = 0.002), and less likely to be of HER2-overexpressed subtype (OR_HER2-overexpressed_ _vs._ _Luminal A(reference)_: 0.89 [0.81–0.98], *p*-trend = 0.018). These associations (odds ratios) were more pronounced among post-menopausal patients. A higher PA score did not improve survival. Higher levels of pre-diagnostic PA were associated with less aggressive tumours in BC patients. This illustrated another benefit of PA in addition to its known role in BC risk reduction.

## 1. Introduction

Breast cancer is the most common cancer among women [1]. Despite the continuous medical advancements to improve detection and intervention methods, it remains important to highlight the major lifestyle factors that can be easily modified and engaged by people in order to reduce breast cancer occurrence [2,3]. One of these factors is physical activity (PA), which has been reported in several epidemiological studies to be a predominant modifiable risk factor for the development of breast cancer [2,4].

Previous studies have demonstrated an inverse association between both pre-diagnostic and post-diagnostic PA with breast cancer mortality [5,6,7,8]. Additionally, the updated World Cancer Research Foundation comprehensive review judged the evidence of physical activity’s preventive role against breast cancer as strong [9]. Furthermore, a dose-dependent relationship between PA and cancer risk and mortality is well established and reported by several meta-analyses and systematic reviews involving various cancers [10,11]. Notably, PA confers an average relative breast cancer risk reduction of 12–25% when comparing individuals with the highest to the lowest category of PA [4,11,12]. To adequately enjoy health benefits, it is recommended that individuals engage in 150 to 300 min of moderate-intensity PA or 75 to 100 min of vigorous intensity aerobic PA per week [13].

Regular PA is associated with many positive impacts on health, well-being and survival in both healthy women and breast cancer patients. However, the relationship between PA and breast cancer severity and aggressiveness such as the stage at diagnosis and grade is not well-known. In this study, we examine the associations between pre-diagnostic PA and breast cancer characteristics at diagnosis and overall survival in a large population of Asian breast cancer patients.

## 2. Methods

### 2.1. Study Population

The Singapore Breast Cancer Cohort (SGBCC) is a multicenter cohort study of breast cancer patients in Singapore. Established in 2010, it was established with the purpose of investigating the associations between various genetic and non-genetic factors and breast cancer risk (cohort profile described in [14]). Patients were recruited across six recruitment sites, between 2010 to 2016, namely: National University Hospital (NUH, Singapore, Singapore), KK Women’s and Children’s Hospital (KKH, Singapore, Singapore), Tan Tock Seng Hospital (TTSH, Singapore, Singapore), National Cancer Centre Singapore (NCCS, Singapore, Singapore), Singapore General Hospital (SGH, Singapore, Singapore), and Changi General Hospital (CGH, Singapore, Singapore). The recruiting hospitals collectively treat ~76% of the breast cancer patients in Singapore [14].

Eligible patients have to be (1) diagnosed with breast carcinoma in situ or invasive breast cancer; (2) citizens or permanent residents of Singapore; (3) aged 21 years and above. As part of the recruitment process, patients completed a structured questionnaire which included questions relating to breast cancer risk factors (i.e., pre-diagnostic PA level, reproductive factors, and family history of breast cancer), with assistance as required from a trained study coordinator. The SGBCC questionnaire was adapted from the KARolinska MAmmography Project for Risk Prediction of Breast Cancer (KARMA) study’s questionnaire [15].

### 2.2. Assessment of PA

Information on PA was obtained from the questionnaire administered at recruitment. Questions included “How much PA/sports do you practice in your free time?” Examples of each PA level were given in the questionnaire (light: walking, driving, housework; moderate: brisk walking, cycling, easy swimming; vigorous: jogging, vigorous swimming, aerobic exercises). Respondents were asked to select the highest level of PA (light, moderate, or vigorous) and to report how much time was engaged in exercise (never, less than one hour per week, one to two hours per week, or more than two hours per week). Information on the highest PA level engaged between the age 18 to 30 (adulthood) was collected.

Based on intensity and duration of PA reported, a PA score ranging from 1 to 5 was calculated for each time point (Figure 1). Patients who engaged in vigorous activities were combined into one group regardless of PA duration due to the small number of patients. As there were very few who chose “never”, these people were classified under Score 1.

### 2.3. Demographics and Breast Cancer Risk Factor Data

Baseline information on sociodemographics and breast cancer risk factors were obtained at the time of recruitment via the structured questionnaire. The variables included ethnicity, smoking (yes, no, or missing) and alcohol consumption (yes, no, or missing), previous benign lump or gynaecological surgery (yes, no, or missing), family history of breast and ovarian cancer (yes, no, or missing), reproductive factors and body size one year prior to diagnosis etc. Body size one year prior to diagnosis was based on the nine-level Stunkard Figure Rating scale of body size, a somatotype pictogram which is validated for the estimation of a person’s body mass index [16,17]. Details on how menopausal status was coded may be found in Appendix A. Medical history, specifically, previous diagnoses of heart attack, asthma, renal disease, stroke, diabetes, and previous cancer, was also collected. Comorbidities were combined and scored according to the Charlson Comorbidity Index (CCI) [18].

### 2.4. Clinical Data

Clinical data on tumour characteristics and treatment modalities were obtained through medical records. The variables included disease stage (stage 0, I, II, III, IV), nodal involvement (yes or no), tumour size (≤2 cm, >2–5 cm, and >5 cm, other/missing), histological grade (well-, moderately-, poorly differentiated), estrogen receptor (ER) status (positive or negative), progesterone receptor (PR) status (positive or negative), human epidermal growth factor receptor 2 (HER2) status (positive or negative), surgery (yes/no), radiotherapy (yes/no), any chemotherapy (neoadjuvant or adjuvant, yes/no), endocrine therapy (yes/no), and targeted therapy (yes/no). Tumour behaviour was derived from stage at diagnosis, where stage 0 indicated non-invasive behaviour, and stages I to IV indicated invasive behaviour. Intrinsic-like subtypes were defined using immunohistochemical markers for ER, PR and HER2 in conjunction with histologic grade; luminal A [ER+/PR+, HER2-, well or moderately differentiated], luminal B [HER2-] (ER+/PR+, HER2-, and poorly differentiated), luminal B [HER2+](ER+/PR+, HER2+, and poorly differentiated), HER2-enriched [HER2+], triple-negative [ER-, PR-, and HER2-] [19].

### 2.5. Passive Follow-Up

Information on vital status and cause of death were obtained via a linkage with the Registry of Births and Deaths [14], using each individual’s unique National Registration Identity Card (NRIC) number. Hospitals had differing schedules in updating their in-house breast cancer registry; in turn, the collection of variables ended at different points (NUH: 30 April 2017; KKH: 30 June 2017; CGH: 16 April 2018; TTSH: 30 April 2018). For SGH and NCCS, not all NRICs are sent to the registry at the same time, the date of follow-up was obtained from the electronic medical records; all recorded deaths are verified with the Registry of Births and Deaths.

### 2.6. Exclusions

Figure 2 summarizes the exclusions performed for this study. We excluded 2 patients without age at diagnosis, 7 male patients, 8 patients without information on PA, and 63 patients diagnosed at below 30 years old. The analytical cohort comprised 7688 breast cancer patients. Further exclusion was performed for survival analysis (refer to Figure 2 for more details), where 6572 patients were retained.

### 2.7. Statistical Analysis

Characteristics of the study population were described by frequency and percentage for categorical variables, and by mean and standard deviation (SD) for continuous variables. The associations between PA and patient characteristics were studied using the Chi-square test and Kruskal-Wallis test, for categorical and continuous variables, respectively.

To assess associations between pre-diagnostic PA score and tumour characteristics, multinomial logistic regression models (*multinom* function in R package “nnet”) were performed, adjusting for age at diagnosis, body size one year prior to cancer diagnosis, ethnicity, and recruitment site. We ran a sensitivity analysis excluding patients who passed away within 1 year of study entry and another separate sensitivity analysis excluding patients who engaged in less than one hour per week of vigorous activity (PA Score 5). We also examined the association of pre-diagnostic PA score and tumour characteristics in subgroups defined by menopausal status at diagnosis (pre- or post-menopausal). Additionally, we investigated the individual effects of pre-diagnostic PA intensity and duration on tumour characteristics. To account for the effect of PA intensity on PA duration and vice-versa, further adjustments were also conducted. Missing data in covariates were replaced by adding a ’missing’ indicator category.

Overall survival was studied using Cox proportional hazard models (survival package in R, where the *Surv (time at entry, follow-up time, event))* command was used to estimate hazard ratios (HR) and corresponding 95% confidence intervals (CI). Time at entry was defined as the time between the date of recruitment and the date of diagnosis. Follow-up time was defined as the time between the date of death/last follow-up date and the diagnosis/recruitment date, censored at ten years post-diagnosis. Breast cancer patients who were recruited ten years after diagnosis were left truncated in the model. In the multivariable Cox regression model, effect of pre-diagnostic PA on survival was adjusted for age at diagnosis and recruitment site. Further adjustments were made for variables found to be significant with survival in univariate models, such as ethnicity, smoking status, CCI, age at menarche, age at first full term pregnancy, hormone receptor use, stage, grade, nodal status, tumour size, estrogen receptor status at diagnosis, and breast cancer treatment history (surgery, chemotherapy, and adjuvant endocrine therapy). Various subgroup analyses were performed to further explore associations between pre-diagnostic PA and survival. These subgroups include hormone receptor and HER2-status, stage at diagnosis, and menopausal status at diagnosis. Further sensitivity analysis excluding patients who engaged in less than 1 hour per week of vigorous activity (PA Score 5) was also conducted.

## 3. Results

Demographics and breast cancer risk factor characteristics of the analytical cohort are described in Table 1a. In these 7688 patients, the mean age at diagnosis was 53.5 years (interquartile range: 46.5–61.2). Table 1b describes disease characteristics and treatment. At diagnosis, 6236 (81.1%) had invasive cancers, 4714 (61.3%) had no nodal involvement; 4957 (64.5%), 4368 (56.8%) and 1490 (19.4%) had ER-, PR- and HER2-positive tumours respectively. Other sociodemographic, medical history, breast cancer risk factors, disease and treatment characteristics that were explored can be found in Appendix A.

### 3.1. Pre-Diagnostic PA and Disease Characteristics

Table 2 shows the associations between pre-diagnostic PA score and disease characteristics, adjusted for age at diagnosis, ethnicity, body size one year prior to diagnosis, and recruitment site. Compared to those with a PA score 2 (reference category for all comparisons), those with highest PA score (score 5) were significantly less likely to be diagnosed with invasive cancer (OR _invasive vs. non-invasive (reference)_ [95%CI]: 0.71 [0.58–0.87], *p* = 0.001, *p*-trend = 0.001). This means that if a patient were to increase her PA score from 2 (reference category) to 5, her risk of developing invasive breast cancer is 0.71 times that of a patient with PA score 2, given the other variables in the model are held constant. With regards to stage, among those with invasive cancer, those who engaged in the highest level of PA were significantly less likely to be diagnosed with stage II breast cancer (OR _stage II vs. stage I (reference)_: 0.80 [0.65–0.98], *p* = 0.028). This inverse association, was, however, not significant when comparing stage I to stages III or IV. Breast cancer patients with higher PA scores were also less likely to be diagnosed with high-grade tumours, especially for those with score 3 (OR _poorly vs. well differentiated (reference)_: 0.69 [0.52–0.93], *p* = 0.014). Other higher PA scores also showed lower risk of developing high-grade tumours but the association was not significant (score 4 OR _poorly vs. well differentiated (reference)_: 0.84 [0.66–1.08], *p* = 0.176; score 5 OR _poorly vs. well differentiated (reference)_: 0.81 [0.62–1.05], *p* = 0.112).

In terms of hormone receptor and HER2 status, higher PA scores were associated with a decreased risk of being diagnosed with ER-negative, PR-negative, and HER2-positive cancers. More specifically, a higher PA score reduced the risk of ER-negative tumours (OR _ER-negative vs. ER-positive (reference)_: 0.94 [0.89–1.00], *p*-trend = 0.049). In addition, compared to those with PA score 2, those with score 4 were less likely to be diagnosed with PR-negative cancers (OR _PR-negative vs. PR-positive (reference)_: 0.82 [0.67–0.99], *p* = 0.041), even though this trend was no longer significant when considering PA score as a continuous variable (*p*-trend = 0.051). Those with the highest PA score had a higher odds of being diagnosed with HER2-negative cancers (OR _HER2-negative vs. HER2-positive (reference)_: 1.29 [1.02–1.62], *p* = 0.033, *p*-trend = 0.002). Notably, an increase in PA score was associated with lower odds of HER2-overexpressed cancer (OR _HER2-overexpressed vs. Luminal A (reference)_: 0.89 [0.81–0.98], *p*-trend = 0.018).

The results did not significantly change in sensitivity analyses excluding patients who passed away within one year of study entry (Appendix A). Similarly, in a separate analysis excluding patients who reported less than 1 hour of vigorous PA (PA Score 5), the results remained largely unchanged, with the exception of ER status (Appendix A). In contrast, PA was not significantly associated with disease aggressiveness among patients diagnosed with non-invasive breast cancer (Appendix A).

### 3.2. Pre-Diagnostic PA and Survival

Among the 6572 patients included for survival analysis, a total of 394 deaths occurred within 10 years after diagnosis. After adjusting for age at diagnosis and recruitment site, higher PA score did not confer any significant survival benefit (HR: 0.98 [0.89–1.08], *p* = 0.647) (Table 3a). Further adjustments for variables that were significant in the univariate Cox regression (Appendix A) did not show any appreciable change (data not shown). Subset analyses by stage, hormone receptor status, and HER2 subsets, adjusted for age at diagnosis and recruitment site, showed similar results (Appendix A). The trends persisted when the outcome considered was deaths due to breast cancer (*n* = 234) (Table 3b). The results remained largely unchanged in sensitivity analysis excluding patients who reported less than 1 hour of vigorous PA (PA Score 5) (Appendix A).

### 3.3. Subset Analysis by Menopausal Status

We conducted further analysis by sub-setting the population according to menopausal status at diagnosis (pre-, *n* = 3617; post-, *n* = 4017) (Table 4). Trends observed among patients with pre-menopausal breast cancer (Table 4a) were less significant than the results observed in Table 2. Even though breast cancer patients with the highest PA score were less likely to develop invasive cancer (OR _invasive vs. non-invasive (reference)_: 0.73 [0.56–0.94], *p* = 0.016, *p*-trend = 0.11), the effects of high PA score on lower stage and grade at diagnosis were no longer significant in this group. Effect of PA score on hormone receptor status (OR _ER-negative vs. ER-positive (reference)_: 0.97 [0.88–1.04], *p*-trend = 0.387; OR _PR-negative vs. PR-positive (reference)_: 0.95 [0.88–1.02], *p* = 0.137) were also no longer significant in this group, even though similar trends were observed. HER2 status remained significantly associated with PA score, where higher PA score reduced the risk of HER2-positive cancers (OR _HER2-negative vs. HER2-positive (reference)_: 1.11 [1.02–1.20], *p*-trend = 0.01).

In contrast, the trends observed between PA score and tumour characteristics among post-menopausal patients (Table 4b) were similar to the results observed in Table 2. Mainly, higher PA score reduced risk of invasive cancer (OR _invasive vs. non-invasive (reference)_: 0.71 [0.50–1.00], *p* = 0.05, *p*-trend = 0.003). Among those with invasive cancer, it was also observed that higher PA score was significantly associated with reduced risk of stage II (OR _stage II vs. stage I (reference)_: 0.71 [0.50–1.00], *p* = 0.048) and high-grade (OR _poorly vs. well differentiated (reference)_: 0.88 [0.79–0.98], *p*-trend = 0.018) cancers. Similar to the pre-menopausal subgroup, effect of PA score on hormone receptor status (OR _ER-negative vs. ER-positive (reference)_: 0.93 [0.85–1.05], *p*-trend = 0.099; OR _PR-negative vs. PR-positive (reference)_: 0.97 [0.89–1.05], *p* = 0.391) was not significant in this subset. Furthermore, the effect of PA score on HER2 status (OR _HER2-negative vs. HER2-positive (reference)_: 1.07 [0.98–1.17], *p*-trend = 0.128) was no longer statistically significant.

Survival benefit was not observed in survival analysis performed with pre- (HR: 1.04 [0.91–1.18], *p* = 0.611) and post-menopausal (HR: 0.91 [0.79–1.05], *p* = 0.213) subgroups (Appendix A).

### 3.4. Subset Analysis by PA Intensity and Duration

In order to study individual effects of PA intensity and duration on disease characteristics, multinomial regression for each variable was conducted separately (Appendix A). Interestingly, increasing intensity and duration of PA have contrasting effects on tumour characteristics. Compared to light levels (reference category), those who engaged in vigorous levels of PA were less likely to develop invasive cancer (OR _invasive vs. non-invasive (reference)_: 0.70 [0.57–0.86], *p* ≤ 0.001). Among those with invasive cancer, those who engaged in higher levels of PA were less likely to develop high-grade (moderate OR _poorly vs. well differentiated (reference)_: 0.80 [0.65–0.97], *p* = 0.024) and HER2-positive (moderate OR _HER2-negative vs. HER2-positive (reference)_: 1.18 [1.00–1.40], *p* = 0.048; vigorous OR _HER2-negative vs. HER2-positive (reference)_: 1.33 [1.06–1.67], *p* = 0.015) tumour. On the contrary, compared to those who engaged in more than 2 h of PA (reference category), those who engaged in only 1 to 2 h of PA per week had a statistically significant decreased risk of high grade (OR _poorly vs. well differentiated (reference)_: 0.73 [0.57–0.94], *p* = 0.014) and large (OR _>5 cm vs. ≤2 cm (reference)_: 0.68 [0.46–1.00], *p* = 0.05) tumour.

When pre-diagnostic PA intensity was further adjusted for pre-diagnostic PA duration, engaging in vigorous levels of pre-diagnostic PA remained significantly associated with reduced risk of invasive cancer (OR _invasive vs. non-invasive (reference)_: 0.71 [0.58–0.88], *p* = 0.001) and HER2-positive tumour (OR _HER2-negative vs. HER2-positive (reference)_: 1.31 [1.04–1.65], *p* = 0.015) (data not shown). However, the effect of pre-diagnostic PA duration on tumour characteristics was no longer statistically significant after adjusting for intensity (data not shown).

## 4. Discussion

In this large, retrospective, multicenter cohort study of 7688 breast cancer patients, a high pre-diagnostic PA score in adulthood (18–30 years) was associated with lower breast cancer stage and grade at diagnosis. High PA score was also statistically associated with ER-positive, PR-positive, and HER2-negative subtypes. We did not observe a significant impact of a high PA score on overall survival. To our knowledge, this is the largest study exploring pre-diagnostic PA and breast cancer characteristics in Asian breast cancer patients.

The influence of PA on breast cancer risk and outcomes has been extensively studied. In these studies, pre-diagnostic PA has been shown to be correlated with a lower incidence of breast cancer [4,11,12]. Conversely, a sedentary lifestyle elevated the risk for cancer development in general [20]. The impact of PA on subgroups by ethnicity and menopausal status has been studied. Compared to Caucasian women, Asians are associated with an appreciable larger reduction in breast cancer risk with PA [12]. While we could not look at breast cancer risk reduction in our study cohort of Asian breast cancer patients, our results suggest that pre-diagnostic PA may affect the aggressiveness of the disease developed. Characteristics that are more favorable were observed in tumours associated with high pre-diagnostic PA scores.

There are inconsistencies regarding the relationship between PA and hormone-receptor status of breast cancers. Our findings in this large Asian breast cancer cohort were in agreement with the majority of previous work. A Spanish case-control study (1389 histologically confirmed invasive BC cases and 1712 controls) found that high PA predisposed to ER-positive, PR-positive and HER2-negative breast cancers [21]. In addition, the Canadian Breast Cancer Study (692 women with incident breast cancer and 644 controls) reported that total lifetime moderate-to-vigorous PA during leisure time exhibited a reduced risk of hormone-receptor (ER or PR) negative breast cancers [22]. The observed effect was confined to HER2-negative tumours [22]. In contrast, a conflicting study found that PA was associated with a reduced risk of ER-positive, PR-positive breast cancers [23]. It should be noted that these studies specified the effects of the type of PA, such as leisure, occupational or household, on hormone receptor and HER2 status in breast cancer [21,22,23], which can explain the difference in findings. However, PA information collected for our study did not allow us to specify types of PA. With regard to the exact relationship between PA and hormone receptors and HER2 status, further investigation will be required.

Stage and grade at diagnosis are important factors determining the aggressiveness of breast cancer. These tumour characteristics impact treatment options, with chemotherapy more likely being offered if the tumour is of a higher stage or grade. The associations between PA and tumour characteristics other than hormone receptors and HER2 status are, however, not commonly reported. Our results regarding invasiveness from stage to hormone receptor statuses highlights the additional roles that pre-diagnostic PA may play, in addition to primary cancer prevention.

Compared to women who screened, non-screeners have been associated with higher rates of advanced and deadly breast cancers [24,25]. In our study, screen-detection is mainly associated with stage 0 and I cancers. Women engaging in physical activity may participate more in screening than physically inactive women. Hence, stage 0 and I breast cancers may be more prevalent among physically active than among physically inactive women. It cannot be discounted that physical activity may be associated with less advanced stage cancer simply because of a difference in screening attendance between physically active and inactive women.

Comparing the highest to the lowest scores of PA, pre-menopausal and post-menopausal women have been reported to be associated with an estimated breast cancer risk reduction of 27%, and 31%, respectively [12]. As menopausal status at breast cancer diagnosis may be associated with distinct risk factors or tumour characteristic profiles, we carried out subset analyses by menopausal status. Our results showed that high PA was associated with more favorable disease at presentation in both subsets, with the association being stronger in the post-menopausal subsets. This aligns with previous publications, where post-menopausal patients were observed to experience greater risk reduction and survival benefit from PA engagement as compared to pre-menopausal patients [26,27]. Chollet-Hinto et al. reported no large differences between menopausal status in a study looking at biologic and etiologic heterogeneity of breast cancer, and concluded that age, rather than menopausal status, may be the key player in determining tumour characteristics [28]. Nonetheless, our results suggest that pre-diagnostic PA is of advantage to both pre- and post-menopausal breast cancer patients.

The mechanisms by which PA reduces breast cancer risk are unclear. Several hypotheses were proposed to explain the effect of PA on cancer development and progression. Particularly for breast cancer, these mechanisms include the lowering of hormones such as estrogen [29] and the avoidance of high insulin levels [30] with PA. Other protective factors include reducing inflammation [31], improving immune system function and decreasing BMI since obesity is linked to many cancers, including breast cancer [32]. As the effects of obesity and PA could be interrelated, the analysis was adjusted for the patients’ body size prior to diagnosis, to exclude obesity as a confounding factor.

Despite the correlations between higher pre-diagnostic PA and more favorable tumour characteristics, there was no significant improvement in overall survival. The potential protective effects of physical activity on survival suggested by other studies could be mediated by physiological changes after diagnosis, such as changes in body size and metabolism [33,34], which was not accounted for in this study. In addition, this study did not account for changes in levels of PA post-diagnosis, which impacts survival [35,36].

Results observed from the separately analyzed effects of PA intensity and duration on disease characteristics were further supported by current guidelines for physical activity, where adults are recommended to engage in either longer duration (150 to 300 min) of moderate or shorter duration (75 to 150 min) of vigorous activity per week [37]. More specifically, it is recommended that individuals perform 500 to 1000 Metabolic Equivalent Tasks (MET) minutes per week to gain potential health benefits or risk reductions. Importantly, our results showed that intensity, compared to duration of PA engagement, is a more important indicator of possible disease outcomes. The exact relationship between PA duration, intensity, and cancer outcome remains unclear. However, the existing literature has shown that compared to moderate-intensity continuous training, engaging in high intensity interval exercise provided greater health improvements such as improved resting blood pressure [38] and cardiorespiratory fitness [39], which might point to the possible effect of PA intensity on improved disease outcomes.

Strengths of this paper included a large study cohort from multiple recruitment sites in Singapore that treat a majority of the breast cancer patients in the country. Participation rate was high (86%) [14] and each questionnaire was completed with little unavailable data. In most cases, there was a dedicated research coordinator to assist the patient in the interpretation of the questionnaire, if needed. The use of a unique National Registration Identity Card (NRIC) for every Singaporean citizen or permanent resident enables healthcare utilisation at all levels to be linked [40]. The clinical data of breast cancer characteristics and outcomes were well kept and retrieved from well-maintained electronic databases, accounting for little missing data. Loss to follow-up due to emigration is expected to be minimal for the duration of the study.

This study is not without limitations. The PA score examined in our study was not a validated tool. Furthermore, patients were limited to the options of light, moderate or vigorous levels with specific durations. Additionally, patients were also asked to report the highest level of PA rather than the habitual pattern of PA. These factors could have resulted in response bias, though there were specific explanations on the questionnaire to guide the patients to make the most appropriate choice. However, accurate measurement of PA is known to be challenging [41]. Nonetheless, our findings can serve as a reference for other studies looking at the associations between PA, breast cancer aggressiveness and fatality. In addition, being a retrospective study, recall bias cannot be eliminated. Moreover, there could be residual confounding and effect modification that could have been missed with our study design. As with all the other epidemiological studies, a causal relationship cannot be conclusively drawn because of various potential confounders. For example, the possibility that the lower cancer stage associated with high PA level may attributed to other factors, such as their engagement in other healthy lifestyle habits, cannot be excluded. To establish a causal effect of PA on breast cancer, large, randomized trials should be planned.

## 5. Conclusions

The relationship between high PA levels and reduced breast cancer risk is recognized. Our study reveals that high pre-diagnostic PA levels may also lower breast cancer aggressiveness in patients who develop the disease. Whether PA levels before breast cancer diagnosis improves survival remains unclear. The results of this study may add value in improving public health recommendations regarding PA for breast cancer risk reduction. Further investigation taking into account more potential confounders, clinical and recommended treatments for breast cancer are needed to clarify the relationship between PA levels and survival.

## Figures and Tables

**Figure 1 cancers-14-01756-f001:**
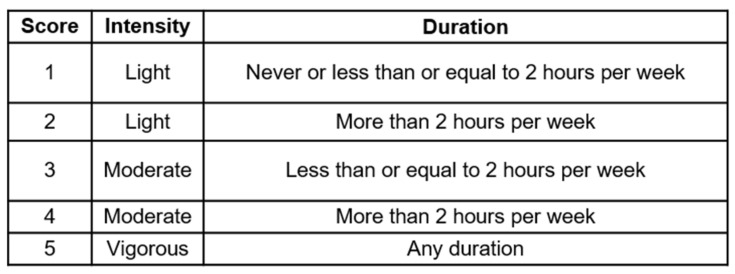
Physical activity score. Scores were calculated based on the highest intensity and duration of physical activity engaged by patient per week.

**Figure 2 cancers-14-01756-f002:**
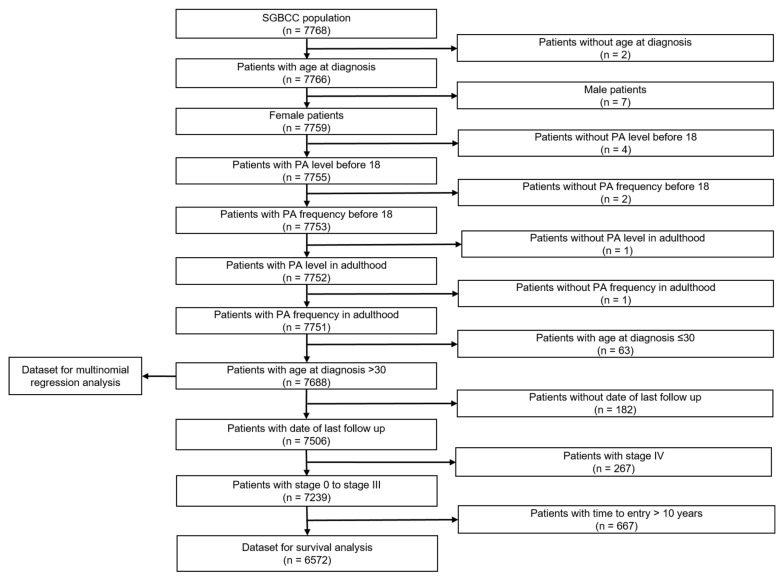
Flow chart of study population. Population comprised of breast cancer patients in the Singapore Breast Cancer Cohort (SGBCC), recruited between 2011 and 2016.

**Table 1 cancers-14-01756-t001:** Characteristics of SGBCC study population by pre-diagnostic physical activity (PA) score (*n* = 7688). *p*-value (*p*) for categorical variables are based on Chi-square test and *p*-value for continuous variables are based on Kruskal-Wallis test.

		Pre-Diagnostic PA Score					
Characteristic	Total, *n* = 7688	1 (Light, <2 h), *n* = 646	2 (Light, >2 h), *n* = 4874	3 (Moderate, <2 h), *n* = 568	4 (Moderate, >2 h), *n* = 801	5 (Vigorous), *n* = 799	*p*-Value
(a) Sociodemographic, medical history, and breast cancer risk factors							
Recruitment site ^a^, *n* (%)							<0.001
NUH	2183 (28.4)	65 (10.1)	1586 (32.5)	64 (11.3)	295 (36.8)	173 (21.7)	
CGH	587 (7.6)	250 (38.7)	134 (2.7)	95 (16.7)	28 (3.5)	80 (10.0)	
KKH	1949 (25.4)	153 (23.7)	1229 (25.2)	154 (27.1)	68 (8.5)	345 (43.2)	
SGHNCC	1544 (20.1)	45 (7.0)	1028 (21.1)	78 (13.7)	278 (34.7)	115 (14.4)	
TTSH	1425 (18.5)	133 (20.6)	897 (18.4)	177 (31.2)	132 (16.5)	86 (10.8)	
Age at diagnosis (years, IQR ^b^)							<0.001
Age at diagnosis (years, IQR)	53.5 (46.5–61.1)	52.0 (45.1–58.8)	55.3 (47.8–62.7)	49.9 (44.2–56.6)	52.1 (45.5–60.4)	49.1 (43.2–55.2)	
Ethnicity, *n* (%)							<0.001
Chinese	6089 (79.2)	554 (85.8)	3882 (79.6)	445 (78.3)	618 (77.2)	590 (73.8)	
Malay	913 (11.9)	57 (8.8)	565 (11.6)	69 (12.1)	100 (12.5)	122 (15.3)	
Indian	439 (5.7)	16 (2.5)	293 (6.0)	33 (5.8)	53 (6.6)	44 (5.5)	
Other	247 (3.2)	19 (2.9)	134 (2.7)	21 (3.7)	30 (3.7)	43 (5.4)	
Body size one year prior to diagnosis, *n* (%)							0.048
Below average	3641 (47.4)	310 (48.0)	2264 (46.5)	293 (51.6)	398 (49.7)	376 (47.1)	
Average	2102 (27.3)	186 (28.8)	1337 (27.4)	148 (26.1)	227 (28.3)	204 (25.5)	
Above average	1930 (25.1)	148 (22.9)	1265 (26.0)	125 (22.0)	175 (21.8)	217 (27.2)	
Missing	15 (0.2)	2 (0.3)	8 (0.2)	2 (0.4)	1 (0.1)	2 (0.3)	
Menopausal status at diagnosis, *n* (%)							<0.001
Pre-menopausal	3671 (47.7)	347 (53.7)	2032 (41.7)	351 (61.8)	413 (51.6)	528 (66.1)	
Post- menopausal	4017 (52.3)	299 (46.3)	2842 (58.3)	217 (38.2)	388 (48.4)	271 (33.9)	
Follow-up duration (years, IQR)							<0.001
Follow-up duration (years, IQR)	5.4 (3.0–9.0)	6.4 (3.7–10.2)	5.1 (2.9–8.6)	6.6 (4.0–10.2)	5.7 (2.8–9.6)	4.8 (2.8–8.3)	
(b) Disease characteristics							
Tumour behaviour, *n* (%)							0.008
Non-invasive	1097 (14.3)	80 (12.4)	672 (13.8)	80 (14.1)	121 (15.1)	144 (18.0)	
Invasive	6236 (81.1)	546 (84.5)	3963 (81.3)	472 (83.1)	644 (80.4)	611 (76.5)	
Missing	355 (4.6)	20 (3.1)	239 (4.9)	16 (2.8)	36 (4.5)	44 (5.5)	
Stage, *n* (%)							0.01
0	1097 (14.3)	80 (12.4)	672 (13.8)	80 (14.1)	121 (15.1)	144 (18.0)	
I	2161 (28.1)	209 (32.4)	1337 (27.4)	170 (29.9)	226 (28.2)	219 (27.4)	
II	2658 (34.6)	230 (35.6)	1705 (35.0)	205 (36.1)	272 (34.0)	246 (30.8)	
III	1150 (15.0)	99 (15.3)	739 (15.2)	76 (13.4)	116 (14.5)	120 (15.0)	
IV ^#^	267 (3.5)	8 (1.2)	182 (3.7)	21 (3.7)	30 (3.7)	26 (3.3)	
Missing	355 (4.6)	20 (3.1)	239 (4.9)	16 (2.8)	36 (4.5)	44 (5.5)	
Grade, *n* (%)							0.149
Well differentiated	1159 (15.1)	94 (14.6)	712 (14.6)	96 (16.9)	141 (17.6)	116 (14.5)	
Moderately differentiated	2766 (36.0)	253 (39.2)	1752 (35.9)	223 (39.3)	275 (34.3)	263 (32.9)	
Poorly differentiated	2834 (36.9)	232 (35.9)	1835 (37.6)	189 (33.3)	302 (37.7)	276 (34.5)	
Missing	929 (12.1)	67 (10.4)	575 (11.8)	60 (10.6)	83 (10.4)	144 (18.0)	
Nodal status, *n* (%)							0.777
Negative	4714 (61.3)	403 (62.4)	2953 (60.6)	367 (64.6)	492 (61.4)	499 (62.5)	
Positive	2556 (33.2)	217 (33.6)	1637 (33.6)	186 (32.7)	251 (31.3)	265 (33.2)	
Missing	418 (5.4)	26 (4.0)	284 (5.8)	15 (2.6)	58 (7.2)	35 (4.4)	
Tumour size (cm), *n* (%)							0.005
≤2	3998 (52.0)	356 (55.1)	2462 (50.5)	325 (57.2)	419 (52.3)	436 (54.6)	
>2–5	2516 (32.7)	212 (32.8)	1623 (33.3)	177 (31.2)	261 (32.6)	243 (30.4)	
>5	399 (5.2)	35 (5.4)	255 (5.2)	29 (5.1)	40 (5.0)	40 (5.0)	
Other/missing	775 (10.1)	43 (6.7)	534 (11.0)	37 (6.5)	81 (10.1)	80 (10.0)	
Estrogen receptor status, *n* (%)							0.469
Positive	4957 (64.5)	430 (66.6)	3108 (63.8)	370 (65.1)	542 (67.7)	507 (63.5)	
Negative	1536 (20.0)	132 (20.4)	1001 (20.5)	107 (18.8)	153 (19.1)	143 (17.9)	
Missing	1195 (15.5)	84 (13.0)	765 (15.7)	91 (16.0)	106 (13.2)	149 (18.6)	
Progesterone receptor status, *n* (%)							0.067
Positive	4368 (56.8)	365 (56.5)	2756 (56.5)	308 (54.2)	489 (61.0)	450 (56.3)	
Negative	2067 (26.9)	193 (29.9)	1317 (27.0)	168 (29.6)	200 (25.0)	189 (23.7)	
Missing	1253 (16.3)	88 (13.6)	801 (16.4)	92 (16.2)	112 (14.0)	160 (20.0)	
HER2 status, *n* (%)							0.006
Positive	1490 (19.4)	169 (26.2)	929 (19.1)	103 (18.1)	158 (19.7)	131 (16.4)	
Negative	3762 (48.9)	310 (48.0)	2380 (48.8)	281 (49.5)	400 (49.9)	391 (48.9)	
Missing	2436 (31.7)	167 (25.9)	1565 (32.1)	184 (32.4)	243 (30.3)	277 (34.7)	
Subtype, *n* (%)							0.014
Luminal A	2294 (29.8)	206 (31.9)	1446 (29.7)	178 (31.3)	240 (30.0)	224 (28.0)	
Luminal B	1018 (13.2)	76 (11.8)	654 (13.4)	69 (12.1)	125 (15.6)	94 (11.8)	
Luminal HER2-like	618 (8.0)	73 (11.3)	374 (7.7)	38 (6.7)	75 (9.4)	58 (7.3)	
HER2 overexpressed	545 (7.1)	70 (10.8)	334 (6.9)	45 (7.9)	49 (6.1)	47 (5.9)	
Triple negative	552 (7.2)	40 (6.2)	364 (7.5)	41 (7.2)	49 (6.1)	58 (7.3)	
Missing	2661 (34.6)	181 (28.0)	1702 (34.9)	197 (34.7)	263 (32.8)	318 (39.8)	

^a^ NUH = National University Hospital, CGH = Changi General Hospital, KKH = KK Women’s and Children’s Hospital, SGHNCC = Singapore General Hospital and National Cancer Centre, TTSH = Tan Tock Seng Hospital. ^b^ IQR = Interquartile range. ^#^ Stage IV cancers were removed in survival analyses.

**Table 2 cancers-14-01756-t002:** Associations between pre-diagnostic physical activity (PA) score and disease characteristics adjusted for age at diagnosis, body size one year prior to diagnosis, ethnicity, and recruitment site (*n* = 7688). Odds ratios (OR) and 95% confidence intervals (CI) were estimated using multinomial regression. *p* indicates *p*-value obtained from the Wald test. Results in bold indicate *p* ≤ 0.05.

Pre-Diagnostic PA Score																	
	1 (Light, <2 h)			2 (Light, >2 h)		3 (Moderate, <2 h)			4 (Moderate, >2 h)			5 (Vigorous)			Continuous PA Score		
	N	OR (95% CI)	*p*	N		N	OR (95% CI)	*p*	N	OR (95% CI)	*p*	N	OR (95% CI)	*p*	N	OR (95% CI)	*p*
Tumour behaviour																	
Non-invasive	80			672	1.00 (Reference)	80			121			144			1097	1.00 (Reference)	
Invasive	546	1.12 (0.86 to 1.47)	0.394	3963		472	1.02 (0.79 to 1.32)	0.867	644	0.93 (0.75 to 1.15)	0.512	611	0.71 (0.58 to 0.87)	**0.001**	6236	0.91 (0.86 to 0.97)	**0.001**
Missing	20			239		16			36			44			355		
Other disease characteristics of patients with invasive cancer (*n* = 6236)																	
Stage																	
I	209			1337	1.00 (Reference)	170			226			219			2161	1.00 (Reference)	
II	230	0.83 (0.66 to 1.04)	0.098	1705		205	0.91 (0.73 to 1.14)	0.407	272	1.03 (0.85 to 1.25)	0.785	246	0.80 (0.65 to 0.98)	**0.028**	2658	0.98 (0.93 to 1.03)	0.352
III	99	0.87 (0.65 to 1.15)	0.332	739		76	0.81 (0.60 to 1.09)	0.171	116	1.01 (0.79 to 1.30)	0.914	120	0.97 (0.75 to 1.24)	0.79	1150	1.00 (0.94 to 1.07)	0.897
IV	8	0.42 (0.20 to 0.89)	**0.023**	182		21	1.07 (0.65 to 1.76)	0.795	30	1.05 (0.69 to 1.59)	0.83	26	1.00 (0.64 to 1.57)	0.998	267	1.07 (0.95 to 1.20)	0.291
Grade																	
Well differentiated	79			570	1.00 (Reference)	80			105			94			928	1.00 (Reference)	
Moderately differentiated	223	1.00 (0.75 to 1.35)	0.982	1507		193	0.88 (0.66 to 1.17)	0.382	230	0.82 (0.63 to 1.05)	0.118	228	0.89 (0.68 to 1.16)	0.375	2381	0.95 (0.88 to 1.01)	0.117
Poorly differentiated	213	0.83 (0.61 to 1.12)	0.216	1619		171	0.69 (0.52 to 0.93)	**0.014**	259	0.84 (0.66 to 1.08)	0.176	242	0.81 (0.62 to 1.05)	0.112	2504	0.94 (0.88 to 1.01)	0.096
Missing	31			267		28			50			47			423		
Nodal status																	
Negative	323			2252	1.00 (Reference)	283			376			342			3576	1.00 (Reference)	
Positive	217	0.92 (0.75 to 1.13)	0.435	1634		184	0.85 (0.69 to 1.04)	0.122	250	0.97 (0.82 to 1.16)	0.752	264	1.00 (0.84 to 1.20)	0.969	2549	1.00 (0.95 to 1.05)	0.99
Missing	6			77		5			18			5			111		
Tumour size (cm)																	
≤2	276			1794	1.00 (Reference)	243			300			292			2905	1.00 (Reference)	
>2–5	212	0.87 (0.71 to 1.08)	0.199	1613		176	0.82 (0.66 to 1.01)	0.066	260	1.04 (0.87 to 1.25)	0.666	243	0.90 (0.74 to 1.09)	0.281	2504	0.99 (0.94 to 1.04)	0.757
>5	34	0.80 (0.52 to 1.22)	0.298	251		28	0.77 (0.50 to 1.18)	0.23	39	0.91 (0.63 to 1.31)	0.613	39	0.88 (0.61 to 1.27)	0.495	391	0.98 (0.89 to 1.08)	0.645
Other/missing	24			305		25			45			37			436		
Estrogen receptor status																	
Positive	379			2722	1.00 (Reference)	322			469			416			4308	1.00 (Reference)	
Negative	122	0.92 (0.72 to 1.17)	0.473	884		98	0.88 (0.69 to 1.13)	0.323	130	0.81 (0.66 to 1.01)	0.057	114	0.83 (0.66 to 1.05)	0.115	1348	0.94 (0.89 to 1.00)	**0.049**
Missing	45			357		52			45			81			580		
Progesterone receptor status																	
Positive	322			2434	1.00 (Reference)	268			429			371			3824	1.00 (Reference)	
Negative	177	1.02 (0.82 to 1.26)	0.883	1156		151	1.06 (0.85 to 1.32)	0.599	167	0.82 (0.67 to 0.99)	**0.041**	156	0.88 (0.71 to 1.08)	0.207	1807	0.95 (0.90 to 1.00)	0.051
Missing	47			373		53			48			84			605		
HER2 status																	
Positive	154			875	1.00 (Reference)	95			145			114			1383	1.00 (Reference)	
Negative	289	0.82 (0.65 to 1.03)	0.093	2247		270	1.24 (0.96 to 1.60)	0.098	382	1.09 (0.88 to 1.34)	0.419	353	1.29 (1.02 to 1.62)	**0.033**	3541	1.10 (1.03 to 1.16)	**0.002**
Missing	103			841		107			117			144			1312		
Subtype																	
Luminal A	190			1375	1.00 (Reference)	174			232			213			2184	1.00 (Reference)	
Luminal B	76	0.81 (0.59 to 1.10)	0.18	635		65	0.80 (0.58 to 1.09)	0.149	120	1.10 (0.86 to 1.40)	0.457	89	0.84 (0.64 to 1.11)	0.223	985	0.99 (0.93 to 1.07)	0.861
Luminal HER2-like	66	1.10 (0.78 to 1.55)	0.587	356		36	0.69 (0.46 to 1.01)	0.059	71	1.05 (0.78 to 1.41)	0.729	53	0.85 (0.61 to 1.19)	0.347	582	0.95 (0.88 to 1.04)	0.265
HER2 overexpressed	63	1.13 (0.80 to 1.61)	0.492	307		40	0.84 (0.58 to 1.23)	0.374	43	0.80 (0.56 to 1.14)	0.222	38	0.73 (0.50 to 1.06)	0.101	491	0.89 (0.81 to 0.98)	**0.018**
Triple negative	39	0.66 (0.44 to 0.99)	**0.046**	342		40	0.79 (0.54 to 1.16)	0.235	45	0.75 (0.53 to 1.06)	0.109	51	0.89 (0.63 to 1.24)	0.487	517	0.98 (0.89 to 1.07)	0.586
Missing	112			948		117			133			167			1477		

**Table 3 cancers-14-01756-t003:** Cox regression model showing association between pre-diagnostic physical activity (PA) score and ten-year survival, adjusted for age at diagnosis and recruitment site (*n* = 6572). Hazard ratios (HR) and 95% confidence intervals (CI) were estimated using Cox regression models.

	Unadjusted		Adjusted	
	HR (95% CI)	*p*	HR (95% CI)	*p*
(a) Overall survival				
Pre-diagnostic PA (continuous)				
	0.93 (0.85 to 1.02)	0.112	0.98 (0.89 to 1.08)	0.647
Age at diagnosis				
	1.03 (1.02 to 1.04)	<0.001	1.03 (1.02 to 1.04)	<0.001
Recruitment site ^a^				
NUH	1.00 (Reference)		1.00 (Reference)	
CGH	0.64 (0.35 to 1.17)	0.146	0.63 (0.34 to 1.16)	0.14
KKH	0.59 (0.46 to 0.76)	<0.001	0.62 (0.48 to 0.80)	<0.001
SGHNCC	1.42 (1.04 to 1.95)	0.028	1.40 (1.03 to 1.92)	0.034
TTSH	0.98 (0.75 to 1.29)	0.893	0.91 (0.69 to 1.20)	0.513
(b) Breast cancer specific survival				
Pre-diagnostic PA (continuous)				
	0.93 (0.83 to 1.05)	0.247	0.96 (0.84 to 1.09)	0.495
Age at diagnosis				
	1.02 (1.00 to 1.03)	0.012	1.01 (1.00 to 1.03)	0.035
Recruitment site				
NUH	1.00 (Reference)		1.00 (Reference)	
CGH	0.00 (0.00 to Inf)	0.992	0.00 (0.00 to Inf)	0.992
KKH	0.57 (0.42 to 0.77)	<0.001	0.58 (0.43 to 0.79)	<0.001
SGHNCC	0.00 (0.00 to Inf)	0.988	0.00 (0.00 to Inf)	0.988
TTSH	0.95 (0.69 to 1.32)	0.777	0.92 (0.66 to 1.27)	0.604

^a^ NUH = National University Hospital, CGH = Changi General Hospital, KKH = KK Women’s and Children’s Hospital, SGHNCC = Singapore General Hospital and National Cancer Centre, TTSH = Tan Tock Seng Hospital.

**Table 4 cancers-14-01756-t004:** Associations between pre-diagnostic physical activity (PA) score and disease characteristics adjusted for age at diagnosis, body size one year prior to diagnosis, ethnicity, and recruitment site for (**a**) pre-menopausal (*n* = 3671) and (**b**) post-menopausal breast cancer (*n* = 4017). Odds ratios (OR) and 95% confidence intervals (CI) were estimated using multinomial regression. *P* indicates *p*-value obtained from the Wald test. Results in bold indicate *p* ≤ 0.05.

Pre-Diagnostic PA Score																	
	1 (Light, <2 h)			2 (Light, >2 h)		3 (Moderate, <2 h)			4 (Moderate, >2 h)			5 (Vigorous)			Continuous PA score		
	N	OR (95% CI)	*p*	N		N	OR (95% CI)	*p*	N	OR (95% CI)	*p*	N	OR (95% CI)	*p*	N	OR (95% CI)	*p*
(a) Pre-menopausal breast cancer (*n* = 3671)																	
Tumour behaviour																	
Non-invasive	49			314	1.00 (Reference)	52			58			98			571	1.00 (Reference)	
Invasive	285	0.98 (0.69 to 1.39)	0.924	1612		288	1.01 (0.73 to 1.41)	0.944	334	1.14 (0.83 to 1.55)	0.42	403	0.73 (0.56 to 0.94)	**0.016**	2922	0.94 (0.87 to 1.01)	0.11
Missing	13			106		11			21			27			178		
Other disease characteristics of patients with invasive cancer (*n* = 2922)																	
Stage																	
I	123			567	1.00 (Reference)	102			112			143			1047	1.00 (Reference)	
II	111	0.68 (0.50 to 0.92)	**0.014**	739		129	0.97 (0.72 to 1.30)	0.846	151	1.09 (0.83 to 1.43)	0.533	171	0.86 (0.67 to 1.11)	0.255	1301	1.02 (0.95 to 1.09)	0.628
III	46	0.84 (0.56 to 1.26)	0.397	255		41	0.88 (0.59 to 1.33)	0.551	60	1.24 (0.87 to 1.77)	0.229	74	1.08 (0.78 to 1.50)	0.634	476	1.06 (0.97 to 1.16)	0.197
IV	5	0.70 (0.26 to 1.90)	0.488	51		16	2.09 (1.11 to 3.94)	0.023	11	1.08 (0.54 to 2.17)	0.82	15	1.29 (0.69 to 2.39)	0.425	98	1.12 (0.95 to 1.33)	0.189
Grade																	
Well differentiated	47			252	1.00 (Reference)	45			51			63			458	1.00 (Reference)	
Moderately differentiated	114	0.99 (0.67 to 1.48)	0.968	615		118	1.09 (0.74 to 1.61)	0.648	117	0.92 (0.64 to 1.32)	0.634	150	0.94 (0.67 to 1.31)	0.716	1114	0.98 (0.89 to 1.07)	0.676
Poorly differentiated	105	0.78 (0.52 to 1.16)	0.22	634		103	0.87 (0.59 to 1.29)	0.482	136	1.05 (0.73 to 1.50)	0.803	159	0.90 (0.64 to 1.25)	0.528	1137	1.00 (0.92 to 1.10)	0.938
Missing	19			111		22			30			31			213		
Nodal status																	
Negative	179			946	1.00 (Reference)	170			195			223			1713	1.00 (Reference)	
Positive	104	0.84 (0.63 to 1.11)	0.222	641		114	0.94 (0.72 to 1.22)	0.627	131	1.04 (0.81 to 1.34)	0.743	177	1.09 (0.87 to 1.37)	0.469	1167	1.04 (0.98 to 1.11)	0.212
Missing	2			25		4			8			3			42		
Tumour size																	
≤2 cm	162			763	1.00 (Reference)	146			151			194			1416	1.00 (Reference)	
>2–5 cm	97	0.72 (0.54 to 0.97)	**0.033**	649		107	0.89 (0.67 to 1.18)	0.426	135	1.10 (0.85 to 1.43)	0.477	165	0.96 (0.76 to 1.22)	0.755	1153	1.03 (0.97 to 1.10)	0.33
>5 cm	21	0.85 (0.49 to 1.49)	0.577	109		18	0.85 (0.49 to 1.48)	0.558	23	1.04 (0.63 to 1.70)	0.886	21	0.74 (0.44 to 1.22)	0.235	192	0.95 (0.84 to 1.09)	0.477
Other/missing	5			91		17			25			23			161		
Estrogen receptor status																	
Positive	197			1092	1.00 (Reference)	207			239			266			2001	1.00 (Reference)	
Negative	62	0.88 (0.63 to 1.24)	0.477	344		47	0.70 (0.49 to 0.99)	**0.046**	68	0.86 (0.64 to 1.17)	0.337	75	0.91 (0.68 to 1.21)	0.502	596	0.97 (0.89 to 1.04)	0.387
Missing	26			176		34			27			62			325		
Progesterone receptor status																	
Positive	179			1012	1.00 (Reference)	176			231			250			1848	1.00 (Reference)	
Negative	80	0.85 (0.62 to 1.16)	0.305	419		77	0.90 (0.67 to 1.22)	0.507	76	0.80 (0.60 to 1.07)	0.137	89	0.80 (0.61 to 1.06)	0.119	741	0.95 (0.88 to 1.02)	0.137
Missing	26			181		35			27			64			333		
HER2 status																	
Positive	78			328	1.00 (Reference)	59			66			71			602	1.00 (Reference)	
Negative	144	0.81 (0.58 to 1.13)	0.206	855		155	1.11 (0.80 to 1.56)	0.531	200	1.21 (0.89 to 1.65)	0.225	226	1.29 (0.95 to 1.75)	0.098	1580	1.11 (1.02 to 1.20)	**0.01**
Missing	63			429		74			68			106			740		
Subtype																	
Luminal A	94			542	1.00 (Reference)	103			116			137			992	1.00 (Reference)	
Luminal B	37	0.89 (0.57 to 1.39)	0.598	219		40	0.94 (0.62 to 1.41)	0.754	64	1.33 (0.94 to 1.89)	0.104	49	0.83 (0.57 to 1.20)	0.329	409	1.01 (0.92 to 1.11)	0.871
Luminal HER2-like	39	1.36 (0.86 to 2.15)	0.189	149		27	0.89 (0.55 to 1.44)	0.638	34	0.99 (0.64 to 1.52)	0.95	41	1.01 (0.67 to 1.50)	0.979	290	0.96 (0.86 to 1.08)	0.507
HER2 overexpressed	29	1.09 (0.65 to 1.83)	0.753	102		21	0.88 (0.52 to 1.51)	0.654	21	0.95 (0.57 to 1.59)	0.846	20	0.71 (0.42 to 1.21)	0.206	193	0.91 (0.80 to 1.04)	0.176
Triple negative	20	0.67 (0.38 to 1.18)	0.162	134		14	0.50 (0.27 to 0.91)	**0.024**	22	0.76 (0.46 to 1.25)	0.285	35	0.95 (0.62 to 1.45)	0.803	225	0.99 (0.88 to 1.11)	0.849
Missing	66			466		83			77			121			813		
(b) Post-menopausal breast cancer (*n* = 4017)																	
Tumour behaviour																	
Non-invasive	31			358	1.00 (Reference)	28			63			46			526	1.00 (Reference)	
Invasive	261	1.38 (0.90 to 2.11)	0.139	2351		184	1.07 (0.70 to 1.63)	0.762	310	0.76 (0.56 to 1.02)	0.068	208	0.71 (0.50 to 1.00)	**0.05**	3314	0.87 (0.80 to 0.96)	**0.003**
Missing	7			133		5			15			17			177		
Other disease characteristics of patients with invasive cancer (*n* = 3314)																	
Stage																	
I	86			770	1.00 (Reference)	68			114			76			1114	1.00 (Reference)	
II	119	1.05 (0.76 to 1.47)	0.753	966		76	0.84 (0.59 to 1.20)	0.331	121	0.96 (0.73 to 1.27)	0.79	75	0.71 (0.50 to 1.00)	**0.048**	1357	0.92 (0.85 to 1.00)	0.06
III	53	0.91 (0.61 to 1.38)	0.662	484		35	0.76 (0.49 to 1.18)	0.215	56	0.86 (0.61 to 1.21)	0.387	46	0.85 (0.57 to 1.26)	0.413	674	0.95 (0.86 to 1.05)	0.286
IV	3	0.24 (0.07 to 0.82)	**0.022**	131		5	0.41 (0.16 to 1.07)	0.069	19	1.04 (0.61 to 1.77)	0.879	11	0.82 (0.42 to 1.60)	0.555	169	1.02 (0.86 to 1.20)	0.822
Grade																	
Well differentiated	32			318	1.00 (Reference)	35			54			31			470	1.00 (Reference)	
Moderately differentiated	109	1.03 (0.65 to 1.62)	0.906	892		75	0.69 (0.45 to 1.07)	0.094	113	0.73 (0.52 to 1.05)	0.086	78	0.85 (0.55 to 1.32)	0.475	1267	0.91 (0.81 to 1.01)	0.069
Poorly differentiated	108	0.88 (0.56 to 1.40)	0.599	985		68	0.54 (0.35 to 0.85)	**0.007**	123	0.70 (0.50 to 1.00)	**0.05**	83	0.73 (0.47 to 1.13)	0.156	1367	0.88 (0.79 to 0.98)	**0.018**
Missing	12			156		6			20			16			210		
Nodal status																	
Negative	144			1306	1.00 (Reference)	113			181			119			1863	1.00 (Reference)	
Positive	113	1.01 (0.75 to 1.36)	0.938	993		70	0.75 (0.55 to 1.04)	0.086	119	0.92 (0.72 to 1.18)	0.508	87	0.88 (0.66 to 1.19)	0.415	1382	0.95 (0.89 to 1.03)	0.195
Missing	4			52		1			10			2			69		
Tumour size (cm)																	
≤2	114			1031	1.00 (Reference)	97			149			98			1489	1.00 (Reference)	
>2–5	115	1.05 (0.77 to 1.44)	0.735	964		69	0.73 (0.53 to 1.02)	0.069	125	0.99 (0.76 to 1.28)	0.926	78	0.80 (0.58 to 1.10)	0.165	1351	0.94 (0.87 to 1.02)	0.124
>5	13	0.66 (0.34 to 1.31)	0.238	142		10	0.62 (0.31 to 1.25)	0.182	16	0.78 (0.45 to 1.36)	0.384	18	1.17 (0.67 to 2.03)	0.575	199	1.02 (0.88 to 1.18)	0.794
Other/missing	19			214		8			20			14			275		
Estrogen receptor status																	
Positive	182			1630	1.00 (Reference)	115			230			150			2307	1.00 (Reference)	
Negative	60	0.96 (0.68 to 1.36)	0.805	540		51	1.23 (0.86 to 1.77)	0.248	62	0.78 (0.58 to 1.05)	0.103	39	0.76 (0.52 to 1.10)	0.14	752	0.93 (0.85 to 1.01)	0.099
Missing	19			181		18			18			19			255		
Progesterone receptor status																	
Positive	143			1422	1.00 (Reference)	92			198			121			1976	1.00 (Reference)	
Negative	97	1.21 (0.88 to 1.64)	0.239	737		74	1.34 (0.96 to 1.86)	0.087	91	0.85 (0.65 to 1.11)	0.23	67	1.00 (0.72 to 1.37)	0.982	1066	0.97 (0.89 to 1.05)	0.391
Missing	21			192		18			21			20			272		
HER2 status																	
Positive	76			547	1.00 (Reference)	36			79			43			781	1.00 (Reference)	
Negative	145	0.84 (0.60 to 1.18)	0.309	1392		115	1.42 (0.95 to 2.12)	0.084	182	0.97 (0.73 to 1.29)	0.825	127	1.26 (0.87 to 1.82)	0.217	1961	1.07 (0.98 to 1.17)	0.128
Missing	40			412		33			49			38			572		
Subtype																	
Luminal A	96			833	1.00 (Reference)	71			116			76			1192	1.00 (Reference)	
Luminal B	39	0.73 (0.47 to 1.14)	0.168	416		25	0.67 (0.41 to 1.09)	0.109	56	0.95 (0.67 to 1.34)	0.768	40	0.91 (0.61 to 1.38)	0.67	576	0.99 (0.90 to 1.10)	0.869
Luminal HER2-like	27	0.77 (0.45 to 1.33)	0.354	207		9	0.41 (0.20 to 0.86)	**0.018**	37	1.13 (0.76 to 1.70)	0.541	12	0.57 (0.30 to 1.09)	0.088	292	0.94 (0.82 to 1.08)	0.366
HER2 overexpressed	34	1.16 (0.72 to 1.88)	0.542	205		19	0.84 (0.49 to 1.46)	0.54	22	0.73 (0.45 to 1.19)	0.203	18	0.80 (0.46 to 1.39)	0.434	298	0.89 (0.78 to 1.02)	0.093
Triple negative	19	0.64 (0.36 to 1.14)	0.131	208		26	1.22 (0.74 to 2.00)	0.442	23	0.77 (0.48 to 1.24)	0.285	16	0.78 (0.44 to 1.37)	0.383	292	0.97 (0.85 to 1.11)	0.706
Missing	46			482		34			56			46			664		

## Data Availability

Due to ethical reasons and institutional guidelines, the data presented in the study cannot be shared publicly. For ethical issues, please contact the National Healthcare Group Domain Specific Review Board (Email: OHRPP@nhg.com.sg) and the SingHealth Centralised Institutional Review Board (Email: irb@singhealth.com.sg). Data are available to interested researchers with some access restrictions applied upon request. All requests can be directed to the Singapore Breast Cancer Cohort (SGBCC) scientific steering committee. Interested researchers may contact the Principal Investigator, Mikael Hartman at mikael_hartman@nuhs.edu.sg for more details. List of available data can be found in https://blog.nus.edu.sg/sgbcc/for-researchers/, accessed on 7 February 2022.

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
