# Peer review of "Associations between Pre-Diagnostic Physical Activity with Breast Cancer Characteristics and Survival"

_cancers, 2022, doi:10.3390/cancers14071756_

Round 1

Reviewer 1 Report

Estimated authors,

The study is interesting, well conducted, and the number of patients is big. On the other side, the methodological work is good but from my point of view authors must correct some important parts of it before the acceptation for publication.

MAJOR REVISION:

- Line 89: Which PA questionary was administered? Please add details. It was validated? If it is, include the reference.

- Lines 102-105 (Figure 1): What is the origin of this score? It was validated? If it is, please add the reference.

In relation with the score, I do not understand in the “score 5” related with strenuous PA, why it is possible to have a score of 5 with “any duration”? So, I understand that if one woman have done 1 minute of vigorous PA, she would have a 5 score?

The WHO recommended 75-100 minutes of vigorous PA each week, then I think it is not possible to control the achievement of this duration with this score. You must modify the PA score to be more accurate, because if not you might have false results and conclusions.

MINOR REVISION:

- Lines 187-193 (Table 1): The table is interesting but too large. I recommend you cut in little pieces or add an epigraph “continue” when the table change to another page, and to include again all the headings of the table (e.g., Characteristic, Total n= 7688, 1 (Light<2h) n=646, …).

- Lines 223-227 (Table 2): Same recommendation that table 1.

- Line 339: “… non-screeners have been been…” The word “been” it is duplicated.

I hope my comments can help to improve your article. Thank you and kind regards

Author Response

Dear Reviewer,

We appreciate the opportunity to submit a revised version of the manuscript. We thank you for your time and comments and have edited the manuscript to address their concerns. A version of the manuscript with changes tracked is uploaded together with this submission.

Estimated authors,

The study is interesting, well conducted, and the number of patients is big. On the other side, the methodological work is good but from my point of view authors must correct some important parts of it before the acceptation for publication.

MAJOR REVISION:

- Line 89: Which PA questionary was administered? Please add details. It was validated? If it is, include the reference.

Our response: PA questions were administered as part of the SGBCC cohort structured questionnaire. The structured questionnaire was adapted from the KARolinska MAmmography Project for Risk Prediction of Breast Cancer (KARMA) study’s questionnaire (doi.org/10.1093/ije/dyw357).

We have added the following clarification to the end of the second paragraph after the “Study population” section in Methods (page 2, line 91 of tracked changes version):

“The SGBCC questionnaire was adapted from the KARolinska MAmmography Project for Risk Prediction of Breast Cancer (KARMA) study’s questionnaire (doi.org/10.1093/ije/dyw357).”

- Lines 102-105 (Figure 1): What is the origin of this score? It was validated? If it is, please add the reference.

Our response: The reviewer is correct in pointing out the score examined in this study has not been validated. We have now highlighted this point as a limitation in the last paragraph of the Discussion section (page 19, line 426 of tracked changes version):

“This study is not without limitations. The PA score examined in our study was not a validated tool. Furthermore, patients were limited to the options of light, moderate or vigorous levels with specific durations. Additionally, patients were also asked to report the highest level of PA rather than the habitual pattern of PA. These factors could have resulted in response bias though there were specific explanations on the questionnaire to guide the patients to make the most appropriate choice. However, accurate measurement of PA is known to be challenging (10.1186/1479-5868-9-103). Nonetheless, our findings can serve as a reference for other studies looking at the associations between PA, breast cancer aggressiveness and fatality.”

In relation with the score, I do not understand in the “score 5” related with strenuous PA, why it is possible to have a score of 5 with “any duration”? So, I understand that if one woman have done 1 minute of vigorous PA, she would have a 5 score?

The WHO recommended 75-100 minutes of vigorous PA each week, then I think it is not possible to control the achievement of this duration with this score. You must modify the PA score to be more accurate, because if not you might have false results and conclusions.

Our response: It was not possible to discern how much time was spent on vigorous PA if the patient responded with less than one hour per week of activity. However, as the SGBCC questionnaire was administered in-person by trained study coordinators, it is likely that a patient would be captured as PA score 5 for short bursts of vigorous PA. We have added results of sensitivity analyses (Supplementary Tables 3 and 7) that excluded 67 patients with less than one hour of PA performed to the manuscript. As expected, the observed odd ratios and p-values remained largely unchanged from the original analysis.

MINOR REVISION:

- Lines 187-193 (Table 1): The table is interesting but too large. I recommend you cut in little pieces or add an epigraph “continue” when the table change to another page, and to include again all the headings of the table (e.g., Characteristic, Total n= 7688, 1 (Light<2h) n=646, …).

- Lines 223-227 (Table 2): Same recommendation that table 1.

Our response: Thank you for the suggestions to the table displays. Some of the fields have been shifted to supplementary material (Supplementary Table 1). We will also work with the journal’s proofing review team to format the tables.

- Line 339: “… non-screeners have been been…” The word “been” it is duplicated.

Our response: The repeated word has been removed.

Reviewer 2 Report

Introduction

lines 68-69: Is there any prior evidence about relationship between PA and BC severity, or do you believe your study is the first?

Methods

Do you have any validity or reliability results for your method of measuring PA? What time period are they asked to recall? 

line 111: what do you mean by body size? Do you mean mass?

line 163-164: What do you mean by "Missingness" at the start of this sentence?

line 170-171: Why did you truncate to 10 years post diagnosis?

line 173: When you write that it was adjusted for site, what is site referring to? Hospital/study site, or side of the body, or something else?

Results

Table 1: in column headers, indicate what the numbers in parentheses are (whether %, SD, etc)

Table 1, first area about site: is the p value column the comparison of total PA between sites or of all PA categories just at the first site (NUH)? Please indicate the answer in the column header. To expand more generally---most of these are multi-level variables. If the p value is for the whole category instead of specifically comparing each level, consider writing the p value in the gray rows that indicate that overall variable

Table 1: Similar to a prior comment, what does body size mean in this study?

line 197: Again, is site the study site or tumor site? Will not point out any further examples in manuscript--please review whole manuscript for clarity of usage of this term.

line 228-229: You observed similar trends in the sensitivity analysis to what? Unclear what things are being compared in this statement

lines 271-273: When you write, "Further survival benefit," do you just mean that there was no significant difference in survival between women diagnosed before and after menopause?

Discussion

line 378: I recommend replacing the word "intense" with "vigorous" to keep with standard guidelines and conventions

Bibliography: check all references for continuity of style

Author Response

Dear Reviewer,

We appreciate the opportunity to submit a revised version of the manuscript. We thank you for your time and comments and have edited the manuscript to address their concerns. A version of the manuscript with changes tracked is uploaded together with this submission.

Introduction

lines 68-69: Is there any prior evidence about relationship between PA and BC severity, or do you believe your study is the first?

Our response: Previous studies between PA and BC either studied prevention (association of BC risk), or outcomes (BC mortality or recurrence). We believe that this is the first study that explores the relationship between PA and BC severity (stage, tumour size, grade at diagnosis etc) in a more in-depth manner in presumably the largest Asian study population. We did an extensive literature search and to the best of our knowledge, the association of certain characteristics, especially grade, with PA was not well studied in the Asian cohort. Hence we have stated at the end of the first paragraph of Discussion (page 17, line 331 of tracked changes version):

“To our knowledge, this is the largest study exploring pre-diagnostic PA and breast cancer characteristics in Asian breast cancer patients.”

Methods

Do you have any validity or reliability results for your method of measuring PA? What time period are they asked to recall? 

Our response: PA questions were administered as part of the SGBCC cohort structured questionnaire, at point of study recruitment (after breast cancer diagnosis). The structured questionnaire was adapted from the KARolinska MAmmography Project for Risk Prediction of Breast Cancer (KARMA) study’s questionnaire (doi.org/10.1093/ije/dyw357). Patients were asked to recall the level of physical activity they engaged between 18 to 30 years old.

We have added the following clarification to the end of the second paragraph after the “Study population” section in Methods (page 2, line 91 of tracked changes version):

“The SGBCC questionnaire was adapted from the KARolinska MAmmography Project for Risk Prediction of Breast Cancer (KARMA) study’s questionnaire (doi.org/10.1093/ije/dyw357).”

We have also highlighted PA score, which is not validated, as a limitation in the last paragraph of the Discussion section (page 19, line 426 of tracked changes version):

“This study is not without limitations. The PA score examined in our study was not a validated tool. Furthermore, patients were limited to the options of light, moderate or vigorous levels with specific durations. Additionally, patients were also asked to report the highest level of PA rather than the habitual pattern of PA. These factors could have resulted in response bias though there were specific explanations on the questionnaire to guide the patients to make the most appropriate choice. However, accurate measurement of PA is known to be challenging (10.1186/1479-5868-9-103). Nonetheless, our findings can serve as a reference for other studies looking at the associations between PA, breast cancer aggressiveness and fatality.”

line 111: what do you mean by body size? Do you mean mass?

Our response: Body size one year prior to diagnosis is based on the nine-level Stunkard Figure Rating scale of body size, a somatotype pictogram which is validated for the estimation of a person’s body mass index (PMID: 6823524, 10.1016/j.jclinepi.2009.08.014). The clarification and relevant references have been added to the manuscript text (under sub-header “Demographics and breast cancer risk factor data”, page 3, line 118 of tracked changes version) for clarity. It has been reported that anthropometric dimensions measured via the pictogram may influence the ability to perform PA (10.1371/journal.pone.0197761).

The decision to choose a visual aided scale over BMI is two-fold. Firstly, BMI was collected at the time of SGBCC recruitment, and hence not fully representative of pre-diagnostic BMI in prevalent cases. Secondly, recall of body build may have an advantage in cases where the patients cannot recall their pre-diagnostic BMI (10.1093/oxfordjournals.aje.a116777).

line 163-164: What do you mean by "Missingness" at the start of this sentence?

Our response: Individuals with missing data in the various categories were included in the analysis. We have changed the phrasing for increased clarity (page 6, line 175 of tracked changes version): “Missing data in covariates were replaced by adding a ’missing’ indicator category.”

line 170-171: Why did you truncate to 10 years post diagnosis?

Our response: The SGBCC study was established in 2010. In view of the maturity of the dataset, the survival analyses were censored at ten years.

line 173: When you write that it was adjusted for site, what is site referring to? Hospital/study site, or side of the body, or something else?

Our response: Site refers to recruitment site. We have changed the phrasing to improve clarity (page 7, line 183 of tracked changes version):”In the multivariable Cox regression model, effect of pre-diagnostic PA on survival was adjusted for age at diagnosis and recruitment site.”

Results

Table 1: in column headers, indicate what the numbers in parentheses are (whether %, SD, etc)

Our response: We have adopted this recommendation.

Table 1, first area about site: is the p value column the comparison of total PA between sites or of all PA categories just at the first site (NUH)? Please indicate the answer in the column header. To expand more generally---most of these are multi-level variables. If the p value is for the whole category instead of specifically comparing each level, consider writing the p value in the gray rows that indicate that overall variable

Our response: We have performed the edits.

Table 1: Similar to a prior comment, what does body size mean in this study?

Our response:  We have added details of the validated nine-level Stunkard Figure Rating scale of body size in the manuscript text.

line 197: Again, is site the study site or tumor site? Will not point out any further examples in manuscript--please review whole manuscript for clarity of usage of this term.

Our response: Site refers to recruitment site (page 10, line 212 of tracked changes version): ”Table 2 shows the associations between pre-diagnostic PA score and disease characteristics, adjusted for age at diagnosis, ethnicity, body size one year prior to diagnosis, and recruitment site.“ We have made the relevant changes for the rest of the manuscript.

line 228-229: You observed similar trends in the sensitivity analysis to what? Unclear what things are being compared in this statement

Our response: We observed similar trends to those in the main study group. We have corrected the phrasing (page 11, line 240 of tracked changes version): “The results did not change appreciably in sensitivity analyses excluding patients who passed away within one year of study entry (Supplementary Table 2).”

lines 271-273: When you write, "Further survival benefit," do you just mean that there was no significant difference in survival between women diagnosed before and after menopause?

Our response: The word “further” has been removed for clarity.

Discussion

line 378: I recommend replacing the word "intense" with "vigorous" to keep with standard guidelines and conventions

Our response: We have adopted the recommendation.

Bibliography: check all references for continuity of style

Our response: We have checked through the references and made the relevant changes.

Round 2

Reviewer 1 Report

Estimated authors,

Thank you for your effort to resolve all my commentaries and suggestions. I hope my comments may have helped to improve your article. I think now the manuscript is ready to publish, congratulations to all your research team.

Thank you and kind regards